

# Class incremental learning of remote sensing images based on class similarity distillation

Mingge Shen[1,2], Dehu Chen[3,4], Silan Hu[5] and Gang Xu[1,2]

[1] Zhejiang College of Security Technology, College of Intelligent Equipment, Wenzhou, Zhejiang, China

[2] Zhejiang College of Security Technology, Wenzhou Key Laboratory of Stereoscopic and Intelligent Monitoring and Warning of Natural Disasters, Wenzhou, Zhejiang, China

[3] Wenzhou University of Technology, College of Architecture and Energy Engineering, Wenzhou, Zhejiang, China

[4] Wenzhou University of Technology, Wenzhou Key Laboratory of Intelligent Lifeline Protection and Emergency Technology for Resilient City, Wenzhou, Zhejiang, China

[5] Macau University of Science and Technology, Faculty of Innovation Engineering, Macau, Macau, China

## ABSTRACT

When a well-trained model learns a new class, the data distribution differences between the new and old classes inevitably cause catastrophic forgetting in order to perform better in the new class. This behavior differs from human learning. In this article, we propose a class incremental object detection method for remote sensing images to address the problem of catastrophic forgetting caused by distribution differences among different classes. First, we introduce a class similarity distillation (CSD) loss based on the similarity between new and old class prototypes, ensuring the model's plasticity to learn new classes and stability to detect old classes. Second, to better extract class similarity features, we propose a global similarity distillation (GSD) loss that maximizes the mutual information between the new class feature and old class features. Additionally, we present a region proposal network (RPN)-based method that assigns positive and negative labels to prevent mislearning issues. Experiments demonstrate that our method is more accurate for class incremental learning on public DOTA and DIOR datasets and significantly improves training efficiency compared to state-of-the-art class incremental object detection methods.

# INTRODUCTION

In various industries such as urban planning, security monitoring, outer space exploration, and many others, remote sensing image processing is widely utilized. It has consistently been a focal point in computer vision due to its high resolution, significant differences in object size distribution within images, and varying orientations. In recent years, the development of deep learning technology has enabled some methods to effectively handle small and multi-directional objects (*Xiaolin et al., 2022*; *Ming et al., 2021*). However, existing methods do not allow for continuous learning of new classes in a human-like manner. In other words, when the model learns a new class, it must retrain with samples from both the previously

Corresponding author
Dehu Chen, chendehu01@163.com

learned class and the new class to achieve satisfactory results. Otherwise, the model will experience catastrophic forgetting. This learning process differs from that of humans. Furthermore, storing samples from old classes consumes a considerable amount of storage space.

For this reason, developing a model that can learn new classes without using old samples and avoid catastrophic forgetting is essential. Some methods attempt to address this issue by updating the parameters of new tasks in the orthogonal space of old tasks from an optimization perspective, thus mitigating forgetting to some extent (*Kirkpatrick et al., 2017*; *Li & Hoiem, 2017*). Other methods (*Rebuffi et al., 2017*; *Rolnick et al., 2019*) adopt a rehearsal mechanism, similar to human review. When learning new tasks, they include a small number of training samples from old tasks. Distillation (*Lee et al., 2019*; *Yang & Cai, 2022*) is widely employed in these methods to ensure the model performs well across all tasks. Yet other methods (*Kirkpatrick et al., 2017*; *Mallya & Lazebnik, 2018*; *Fernando et al., 2017*) are based on the over-parameterized characteristics of deep neural networks, activating or expanding neurons for different tasks. However, these methods lack the utilization of learned knowledge, akin to humans reviewing old knowledge to better learn new knowledge. Furthermore, recent work (*Simon et al., 2022*) employs Mahalanobis similarity as a learning parameter to learn meaningful features, but it still encounters the issue of linearly increasing the number of parameters as the number of tasks increases. Most existing lifelong learning methods assume that tasks originate from the same distribution, ignoring the more general situation where tasks come from different domains.

There are also incremental object detection methods designed to address catastrophic forgettings, such as *Liu et al. (2020a)*, which restricts the updating of weights on new classes based on the importance of the impact of a new class on the model and limits the update of weights on new tasks. A regularization term is introduced to constrain the update of model weights on a new class. With a certain number of neurons added to the model to learn the new class, *Dong et al. (2021)* and *Shieh et al. (2020)* ensure that the model learns the new class while maintaining the model's parameters for the old classes simultaneously. In *Hao, Fu & Jiang (2019a)*, distillation techniques are employed to ensure that the network model remembers the old classes while learning a new one. *Shieh et al. (2020)* use a replay-based approach, *i.e.,* storing some representative samples of the old classes, and acquiring new knowledge by using new task samples and the stored old samples. However, there are two main problems with existing methods:

1. The existing methods cannot fully exploit the similarity information among classes as humans can. For instance, humans can learn to detect helicopters faster in a model that has learned to detect aircraft.

2. With the increase in classes, a larger model, storage and computational costs will be inevitable, and the model's accuracy will decrease rapidly.

To deal with the above issues, the main contributions of this article are concluded as follows:

1. Based on class similarity distillation, we propose a method for class incremental object detection, which can dynamically adjust the distillation weights according to the similarity between new and learned classes, *i.e.,* if the new task is more similar to the

old class. In that case, the distillation weights on the new class can be increased to enhance the forward transfer ability of the model and vice versa to ensure the unity of model plasticity and stability.

2. By maximizing the mutual information between the new class and the old task, we propose a global similarity loss (GSL) that maximizes the extraction of similarity information between the new and old classes.

3. The experiments demonstrate that our model can guarantee high accuracy without adding additional storage or computing resources.

The related work is briefly reviewed in the "Related work" section, and the proposed approach is clarified in the "Methods" section. Experiments and implementation details are provided in the "Results" section to validate our method's effectiveness using two standard remote sensing datasets. There is further discussion of the article's shortcomings in the "Discussion" section, and a conclusion is in "Conclusions".

## RELATED WORK

In recent years, deep learning-based object detection methods have seen rapid development. Generally, these methods can be classified into two categories: anchor-based, such as the R-CNN series (*Girshick, 2015*; *Ren et al., 2015*) and YOLO series (*Redmon et al., 2016*; *Redmon & Farhadi, 2017*; *Redmon & Farhadi, 2018*), and anchor-free, which are not based on preset anchors, such as FCOS (*Tian et al., 2019*) and DETR (*Zhu et al., 2020*). Both algorithms are highly accurate in detecting objects, but they cannot handle class incremental learning tasks. In recent years, some class incremental object detection algorithms (*Yang et al., 2022*; *Zhang et al., 2021*; *Ul Haq et al., 2021*) have emerged that can incrementally learn new tasks. These methods are divided into three main categories: parameter isolation-based, replay-based, and regularization-based.

The first category is the rehearsal-based method, similar to human review. When the model learns new tasks, the impact of old tasks is considered simultaneously, allowing the model to better remember old tasks and avoid catastrophic forgetting. This method widely uses distillation technology, as it can quickly learn new tasks with few samples. The most representative is the ICARL algorithm (*Rebuffi et al., 2017*), which uses a teacher network and student network to enable all learned tasks to converge quickly with a small number of training samples. Therefore, only a small number of previous task samples need to be stored when learning a new task. To save memory overhead, *Rolnick et al. (2019)* propose reservoir sampling to limit the number of stored samples to a fixed budget data stream. Continual prototype evolution (CPE) (*De Lange & Tuytelaars, 2021*) combines the nearest-mean classifier approach with an efficient reservoir-based sampling scheme. More detailed experiments on the rehearsal for lifelong learning are provided in (*Masana et al., 2020*).

Compared to directly storing samples, another representative method is GEM (*Lopez-Paz & Ranzato, 2017*). It stores the gradient of previous tasks instead of training samples, ensuring the direction of the gradient update for new tasks is orthogonal to the previous tasks, reducing interference with prior knowledge. Many methods adopt similar principles.

To further save memory space, numerous GAN-based methods are proposed to generate high-quality images and model the data-generating distribution of previous tasks, retraining on generated examples (*Robins, 1995*; *Goodfellow et al., 2014*; *Shin et al., 2017*; *Ye & Bors, 2021*). Although GAN-based methods reduce storage space, they introduce many additional calculations.

The second category is the regularization-based method. The main idea of these methods is to add a regularization term of parameter importance, which can reduce the update of essential parameters for old tasks and increase the update of unimportant parameters. To evaluate the importance of parameters, LwF (*Li & Hoiem, 2017*) limits the update of parameters according to the difference between the new task and the old task. EWC (*Kirkpatrick et al., 2017*) determines the importance of weight parameters according to the training Fisher information matrix. However, with increased tasks, Fisher regularization will excessively limit the network parameters, resulting in the inability to learn more new tasks. To address this problem, some methods, such as the SI algorithm (*Zenke, Poole & Ganguli, 2017*), determine the importance of network parameters according to the variation range of network parameters from old tasks to new tasks. However, the parameter update method of random gradient descent often makes the results unstable. In contrast, MAS (*Aljundi et al., 2018*) allows importance weight estimation to provide datasets without supervision, enabling it to perform user-specific data processing. Variational continuous learning (VCL) (*Nguyen, Ngo & Nguyen-Xuan, 2017*) uses a variational framework for continuous learning.

Some Bayesian-based works (*Ahn et al., 2019*; *Zenke, Poole & Ganguli, 2017*) estimate the importance of weights online during task training. *Aljundi et al. (2018)* propose an unsupervised parameter importance evaluation method to increase flexibility and online user adaptability. Further work by *Lange et al. (2020)* and *Aljundi, Kelchtermans & Tuytelaars (2019)* extends this method to the case of no task setting. However, these methods are generally difficult to converge.

The third category is neuron activation or expansion methods, which activate different parameters of the network for different tasks or add additional parameters for new tasks in advance if the deep neural network is over-parameterized. However, the increased number of tasks can easily lead to the saturation of model parameters.

PackNet (*Mallya & Lazebnik, 2018*) prunes weights in the network according to the importance of the weights. Only the first 50% of the weight is selected each time to train the current task. HAT (*Serra et al., 2018*) either freezes previous task parameters or dedicates a model copy to each task when learning new tasks. Alternatively, the architecture remains static, with fixed parts allocated to each task. The previous task parameters are masked during new task training, and each task feature is converted into an embedding. After passing these embeddings, the network converts them into masks. HAT (*Serra et al., 2018*) takes sparsity as the loss function, which is more intelligent. These works typically require a task oracle, activating corresponding masks or task branches during prediction. Therefore, they are restrained to a multi-head setup, incapable of coping with a shared head between tasks. Expert gate (*Aljundi, Chakravarty & Tuytelaars, 2017*) avoids this problem by learning an auto-encoder gate.

Compared to fixed network weight numbers, there are also some methods such as progressive networks (*Rusu et al., 2016*), dynamic memory networks (*Perkonigg et al., 2021*), and DER (*Yan, Xie & He, 2021*) that increase the network structure. Whenever a new task is performed, appropriate neurons are added to train the new task. However, these methods cannot be used for large-scale task learning due to the limitation of the number of parameters.

In recent years, several works in remote sensing have been using incremental learning methods to detect optical remote sensing images acquired through remote sensing, SAR, and hyperspectral images as a result of the above methods of incremental object detection. These works have been achieving some results by using these incremental learning methods. Although remote sensing image object detection is a complex task, studies have yet to be conducted on class incremental object detection owing to its high complexity. Instead of adapting to unseen new classes, acquiring new samples from old classes will improve the detector rather than adapting to unseen new classes. The article's authors propose a class incremental learning method based on multiscale features to detect objects in more than one direction. *Dong et al. (2021)* proposed a method that could reduce the number of new classes by using a class incremental learning method that combines a teacher-student structure and selective distillation to reduce the number of new classes.

In *Li et al. (2022)*, a Rank-aware Instance Incremental Learning (RAIL) method, based on the notion of a rank-aware instance incremental learning measure, is proposed. RAIL considers the differences between learning values in data learning order and training loss weights. Rank scores are then used to weigh the training losses to balance the learning contributions. However, existing research on continual object detection is still in its early stages, and current approaches primarily fall into two main categories: experience replay (*Joseph et al., 2021a*) and knowledge distillation (*Liu et al., 2020b*; *Shmelkov, Schmid & Alahari, 2017*). *Joseph et al. (2021a)* stores representative examples in memory, allowing them to be trained alongside new category samples and fine-tuning the model. *Shmelkov, Schmid & Alahari (2017)* employs knowledge distillation for both object localization and classification. *Liu et al. (2020b)* further utilizes attentive feature distillation to extract essential knowledge through both top-down and bottom-up attention mechanisms.

However, when the distribution of the new class is very different from the distribution of the old class, existing methods based on knowledge distillation cannot effectively learn the information of the new class. Furthermore, even though complex models can be used to increase the detection accuracy of individual tasks, it is detrimental to knowledge distillation when this happens. Based on human learning, the efficiency of learning increases as more knowledge is learned since humans can use the learned similarity information to increase the speed of learning.

Inspired by human learning behavior, we propose a new method to continuously detect objects in remote sensing images by considering the similarity and differences between new and old classes by utilizing knowledge distillation to its fullest extent. As a result, the efficiency of the model can improve as more knowledge is learned.

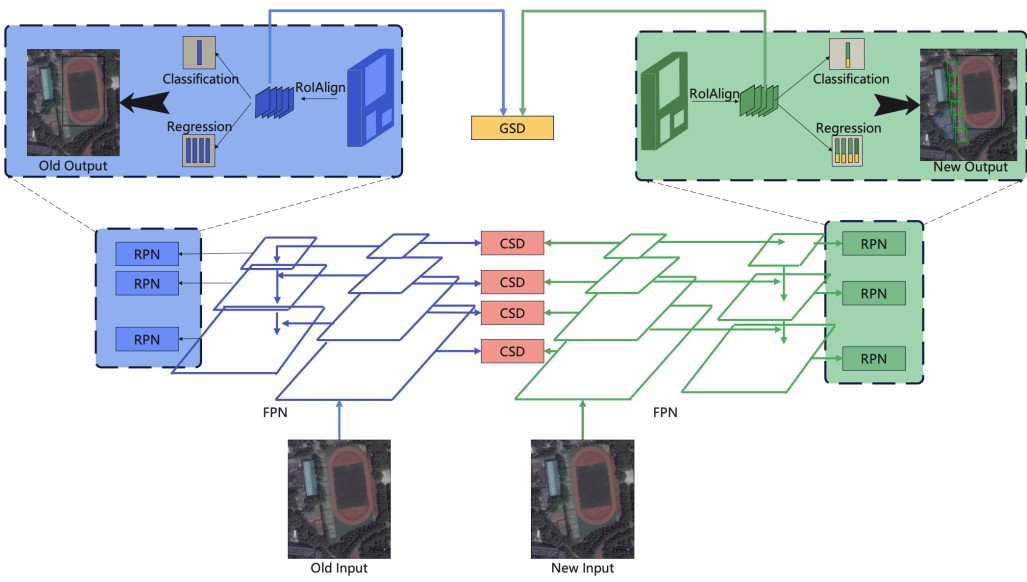

**Figure 1** **The framework of proposed method, We use the faster R-CNN detection framework with the backbone of FPN.** To maximize the similarity between learning tasks, we use class similarity distillation (CSD) loss in the block-wise level and global similarity distillation loss in the instance level.

## METHODS

Our proposed class incremental object detection framework is shown in Fig. 1. We use the Faster R-CNN detection framework with the backbone of the feature pyramid network (FPN) (*Lin et al., 2017*). To maximize the similarity between learning tasks, we use class similarity distillation (CSD) loss at the block-wise level and Global Similarity Distillation loss at the instance level. In addition, we use an RPN-based method to assign positive and negative labels to prevent the mislearning problem caused by the new class being taught against the background of the previous class.

### Problem setting

Our class incremental learning setup is as follows, given an object detector that has been trained on $C$ classes, when a new class $C_n$ comes and we are given a dataset $D_n$ which comprises a set of pairs $(X_n, Y_n)$, where $X_n$ is an image of size H × W and $Y_n$ is the ground-truth. Here, $Y_n$ only consists of labels in current classes $C_n$. The model should be able to predict all classes $C_1$: $C_n$ in the history.

### Class similarity distillation

The detail of the proposed CSD is shown in Fig. 2. When learning a new class. We train the new model using the new class samples and labels, consider the output of new samples in the old model, and ensure that the new model avoids catastrophic forgetting. In order to avoid the instability caused by large models, we use the CSD at the block level. The proposed CSD can make better use of similar information. After each block, we use the weighted distillation loss to decide the degree of distillation according to the similarity

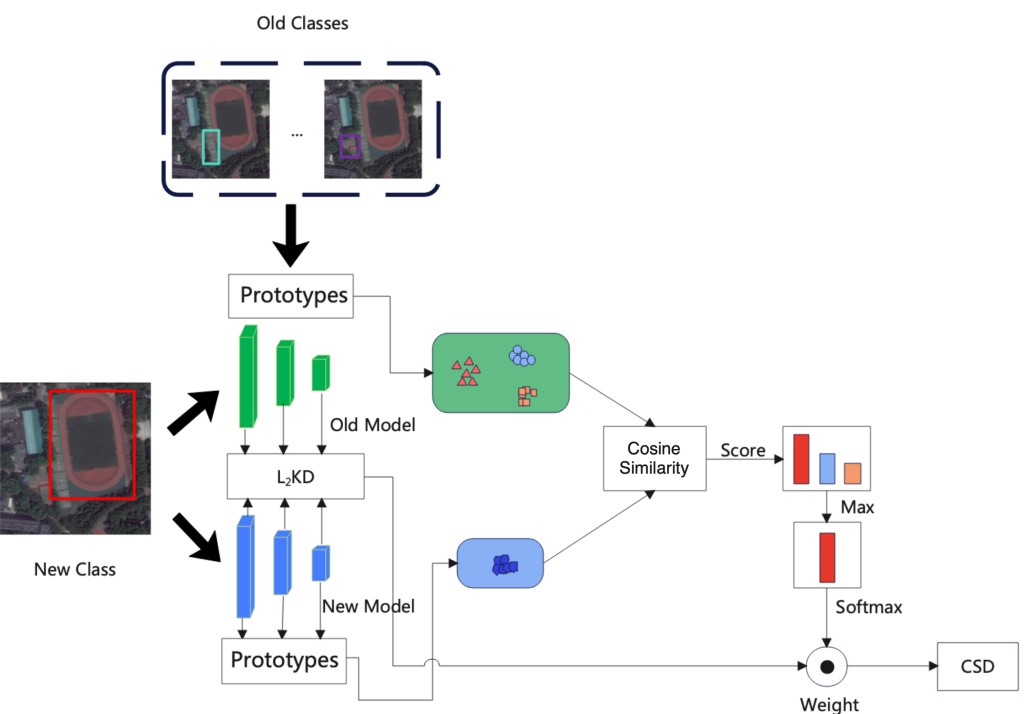

**Figure 2** The detail of proposed class similarity distillation (CSD).

between the new class and the old classes, *i.e.*, if the old classes are more similar to the new classes, then the weights are small in the distillation process of the new class, and vice versa.

We first obtain the prototype of new class k by computing an in-batch average shown in Fig. 2 on $Z = R^{H \times W \times C}$. Given a batch of feature maps $B = R^{B \times H \times W \times C}$, we flatten the batch, height and width dimensions and index the as $z_i$, where $i = 1, \ldots, BHW$. The centroid of class $c$ is computed as Eq. (1)

$$p_k = \frac{\sum_{i=1}^{BHW} z_i 1\left[y_i = k\right]}{k} \tag{1}$$

where $1[y_i = k] = 1$ if the label $y_i$ is k, otherwise $1[y_i = k] = 0$. The cummulative prototypes $P_1 : P_k$ of all classes from class 1 to class t are computed at the end of class k.

We construct a prototype map $m_x = R^{H x W \times C}$ where each pixel $x$ contains a prototype vector $m_x = p_k$ Then we compute a similarity map $S = R^{H x W x V}$ between the prototype $m_x$ of a new class in each pixel $x$ and the prototype is $p_k$ of old class. Each entry $S_{(x,k)}$ is cosine similarity between m and $p_k$, the normalized similarity map S is defined as

$$S = \frac{\exp\left(\frac{m_x \cdot p_k^{t-1}}{\|m_x\| \cdot \|p_k^{t-1}\|}\right)}{\sum_{j=1}^{k} \exp\left(\frac{m_x \cdot p_j^{t-1}}{\|m_x\| \cdot \|p_j^{t-1}\|}\right)}. \tag{2}$$

Finally the class similarity distillation loss distills the weighted outputs of the old model and the new model:

$$Ls(k,x) = \sum_{k=1}^{m} S(x-k)^2 \qquad (3)$$

where $k$ and $x$ are old class and new class features.

Learning this similarity provides two benefits. As a first step, the model can relate the new class to what it had previously learned, which facilitates the transfer of the old knowledge to the new class for a better learning experience. Second, it encourages the model to learn the underlying class hierarchy implicitly. We do not need to save the class ID and only save the prototype when the new class are well trained so that we can learn the similarity of the new class more quickly.

## Global similarity distillation

In order to maximize the extraction of correlation features of different class objects in remote sensing images, we propose the global similarity loss (GBL) to maximize the similarity information of old class and new new by maximizing the mutual information in the instance level before classification and regression results. The GBL is shown in Eq. (4)

$$L_g = \frac{S(x_t, y_t)}{\sum_{j=1}^{k} S(x_j, y_t)} \qquad (4)$$

where $x_t$ is the old instance-level class feature, and $y_t$ is current instance-level class feature, $x_j$ is noisy old class feature. and $S()$ is cosine similarity. Maximizing this equation is equivalent to maximizing the relationship between the model discriminated learned and unlearned classes, and maximizing the mutual information of the new class and the old classes.

## Positive and negative samples assignment based on RPN

In general, the way *Ren et al. (2015)* assigns positive and negative labels for training samples in remote sensing datasets is based on the size of anchors. Some datasets contain multiple class samples simultaneously. Thus, some unknown class positive samples are labeled as new class negative samples, leading to decreased efficiency and accuracy in learning these samples.

To solve this problem, we propose an RPN-based technique for assigning positive and negative samples to label potential new classes. Specifically, these new classes will be designated as unknown samples, which means they will not be included in the training of positive and negative samples, thereby avoiding the problem of new tasks appearing in old tasks, which would result in inadequate training.

Firstly, based on the characteristics of the region proposal network (RPN), which can output the class probability scores and the bounding boxes of almost all objects, our approach is to treat those objects with higher objective scores but do not have higher IoU with ground-truth scores as potential unknown objects that should not be included in the training of the positive and negative samples. Specifically, a negative sample is defined as "1" where the probability score ranking of the last k objects is less than a certain threshold, and at the same time, the IoU of the ground-truth is less than a certain threshold.

## Loss function

The loss function of the entire framework is shown as Eq. (5).

$$L = L_{det} + aL_s + bL_g. \tag{5}$$

The first of them is the faster R-CNN detection loss function, the second term is the proposed class similarity distillation loss, and the third is the global similarity loss. We use gradient descent with momentum to optimize the model. During the training period, we first fix the other parameters and train the RPN of new class branches of the parameters to converge, and then we train all the parameters. The results prove the effectiveness of the training method.

## RESULTS

We used two public remote sensing datasets, DOTA (*Xia et al., 2018*) and DIOR (*Li et al., 2020*), to verify the effectiveness of the proposed method; first, we compared with some State-of-the-Art (SOTA) methods, and then we conducted an ablation study to verify the effectiveness of the proposed two distillation loss functions. The specific training parameters were set as follows; we cropped the image to $800 \times 800$ size, the batch size was set to 2, the momentum was set to 0.9, the iteration, the number of times, was set to 50,000, the initial learning rate was set to 0.0025, every 10,000 times was reduced to one-tenth of the original, IoU was marked as the correct result when it was significant with 0.7, the RPN output was 128 for both positive and negative samples, and the experiments all used horizontal bounding boxes.

There are 2,826 images in the DOTA dataset and 188,282 instances with image sizes ranging from $800 \times 800$ to $4000 \times 4000$, containing 15 classes, and we use the first eight classes as old classes. We incrementally learn the other seven classes.

There are 11,738 images in the DIOR dataset, and 20 classes contain 190,288 instances. We set the first ten classes are old classes and the last ten classes are new.

## Evaluation criteria

To obtain a generic model performance estimate, after training task t, we compute the average accuracy (AA) on all testing datasets of tasks T. The average accuracy is defined as Eq. (6). The higher the average accuracy, the better the performance of the model.

$$AA = \frac{1}{T} \sum_{t=1}^{T} \left( \frac{\text{TP}_t + \text{TN}_t}{\text{P}_t + \text{N}_t} \right) \times 100 \tag{6}$$

where TP and TN are the numbers of correctly classified samples. $P_t$ and $N_t$ are the number of positive and negative samples for task t. T is the total number of tasks.

## Performance evaluation

We used ResNet as the uniform backbone, and it can be seen from the AA on both datasets in Table 1 that the proposed method improves by 5% compared with the SOTA method FPN-IL (*Chen et al., 2020*). This is because our method can consider the old class features when learning new classes, thus obtaining a higher AA. Other methods use traditional

**Table 1  The detection results (AA%) of all six compared methods.**

| Method | Basic Archietcture | Learning Stategy | DOTA AA% | | DIOR AA% | |
|---|---|---|---|---|---|---|
| | | | Old 8 | New 7 | Old 10 | New 10 |
| Fast-IL | | Incremental | 26 | 13 | 31 | 19.4 |
| | | Joint-training | 26.8 | 22.6 | 33.5 | 34.1 |
| Faster-IL | | Incremental | 36.1 | 26.4 | 36.7 | 47.0 |
| | | Joint-training | 41.5 | 26.9 | 47.4 | 47.7 |
| FPN-IIL | | Incremental | 69.2 | 60.7 | 68.8 | 68.1 |
| | | Joint-training | 69.8 | 60.8 | 69.4 | 71.3 |
| Meta-ILOD | | Incremental | 70.1 | 61.6 | 69.7 | 68.1 |
| | ResNet | Joint-training | 69.8 | 70.5 | 69.9 | 72.3 |
| SID | | Incremental | 69.6 | 61.3 | 72.1 | 70.9 |
| | | Joint-training | 70.3 | 71.9 | 70.3 | 72.1 |
| ORE | | Incremental | 69.8 | 61.6 | 69.7 | 71.1 |
| | | Joint-training | 70.9 | 72.4 | 70.5 | 72.2 |
| CWSD | | Incremental | 68.7 | 60.5 | 68.2 | 70.5 |
| | | Joint-training | 71.7 | 73.6 | 71.8 | 73.5 |
| CSD(Ours) | | Incremental | 70.5 | 62.7 | 70.5 | 72.4 |
| | | Joint-training | 71.7 | 73.6 | 71.8 | 73.5 |

methods to generate class agnostic RoI or use the dispersion of features before RPN to learn new knowledge and do not fully use the new class information of similarity, so the detection results are unsatisfactory.

Table 1 shows the detection results on each class in the new seven classes of the DOTA dataset. The detection result by Fast-IL (*Shmelkov, Schmid & Alahari, 2017*) is poor in detecting every class, as the detection framework is not effective. The Faster-IL (*Hao et al., 2019b*) and FPN-IL (*Chen et al., 2020*) are much better than Fast-IL, but the average accuracy (AA) is lower as the number of classes increases. Meta-ILOD (*Joseph et al., 2021b*) uses meta-learning to learn a global optimum solution without learning the similarity between classes. SID (*Peng et al., 2021*) employs distillation in some intermediate features, while our method performs global information distillation at various scales, resulting in better performance compared to SID. The training process of ORE (*Joseph et al., 2021a*) is more complicated, requiring a long pre-training period to achieve good results. Compared with the CWSD (*Feng et al., 2021*), the proposed method is supplemented by weighted similarity not only supplements similar features. The proposed method has improved approximately 1% on AA compared to the four most recent methods, and as the classes increase, the detection of the new class does not show a noticeable drop.

To demonstrate in more detail that the proposed method can learn the similarity information among classes well, we list the average accuracy of each class for each class, as shown in Table 2. In the DOTA dataset, because the class of baseball field (BF) was learned before when learning new categories such as tennis court (TC) and basketball court (BC), which have relatively similar characteristics to a baseball court (BC), the accuracy of our method in detecting these is significantly higher than that of other methods. Since

**Table 2  The detection result (AA%) on each class in new seven classes of DOTA dataset.**

| Method | Basic Architecture | Learning Stategy | DOTA (New seven Classes) AA% | | | | | | |
|---|---|---|---|---|---|---|---|---|---|
| | | | BC | ST | SBF | TR | Harbor | SP | HC |
| Fast-IL | | Incremental | 19.2 | 17.6 | 28.8 | 19.4 | 17.4 | 13.2 | 4.7 |
| | | Joint-training | 26.8 | 22.6 | 33.5 | 34.1 | 36.7 | 20.3 | 18.2 |
| Faster-IL | | Incremental | 36.1 | 26.4 | 36.7 | 47.0 | 42.4 | 35.4 | 9.7 |
| | | Joint-training | 41.5 | 26.9 | 47.4 | 47.7 | 45.1 | 36.1 | 8.1 |
| FPN-IL | | Incremental | 69.2 | 60.7 | 68.8 | 68.1 | 70.6 | 62.7 | 45.9 |
| | | Joint-training | 69.8 | 60.8 | 69.4 | 71.3 | 74.2 | 62.6 | 35.7 |
| Meta-ILOD | ResNet | Incremental | 69.7 | 60.4 | 69.7 | 69.3 | 71.7 | 63.8 | 45.9 |
| | | Joint-training | 70.3 | 61.3 | 69.5 | 70.8 | 74.2 | 63.3 | 37.1 |
| SID | | Incremental | 69.6 | 60.5 | 69.6 | 69.3 | 70.4 | 63.8 | 46.9 |
| | | Joint-training | 70.4 | 62.4 | 70.8 | 71.4 | 75.5 | 62.7 | 36.3 |
| ORE | | Incremental | 69.4 | 60.4 | 69.1 | 68.1 | 70.6 | 62.7 | 45.9 |
| | | Joint-training | 70.1 | 61.5 | 69.8 | 71.9 | 74.5 | 62.1 | 36.2 |
| CWSD | | Incremental | 68.4 | 60.2 | 68.6 | 69.2 | 71.3 | 64.2 | 46.8 |
| | | Joint-training | 69.8 | 60.8 | 72.4 | 71.3 | 76.9 | 63.8 | 37.2 |
| CSD(Ours) | | Incremental | 69.5 | 60.7 | 69.1 | 70.4 | 72.5 | 65.3 | 46.8 |
| | | Joint-training | 69.8 | 60.8 | 72.4 | 71.3 | 76.9 | 63.8 | 37.2 |

our approach uses the same backbone architecture as FPN-IL, it has similar performance during joint training without having learned from similar samples. However, due to our method's ability to fully learn similar information, it performs better when learning from similar samples later on, such as SBF, SP, HC, etc. Meta-ILOD (*Joseph et al., 2021b*) employs meta-learning to obtain a global optimum solution without learning inter-class similarities, while our approach conducts global information distillation at multiple scales, leading to enhanced performance in comparison. The training process of ORE (*Joseph et al., 2021a*) is complex, and the CWSD (*Feng et al., 2021*) is not in line with the continual learning setting. Therefore, the proposed method achieves roughly a 1% average improvement in AA compared to the four most recent techniques mentioned above. Although the accuracy of each class varies slightly with the learning order, the overall AA and joint training are comparable due to the learning of the old class similarity by the proposed method, and there is a significant improvement in AA when the similarity task is learned later. This shows that the proposed method is stable and effective.

Figure 3 shows the visualization detection results of the proposed method on the DOTA dataset with the truck as the old task to learn the new task sedan, and the visualization detection results with the soccer ball field (SBF) as the old task to learn the basketball court (BC) and tennis court (TC). From the detection results, we can see that our method obtains high average accuracy on both new and old classes. In contrast, other methods have many missed detections on the old class, as shown in the red box, which is because our

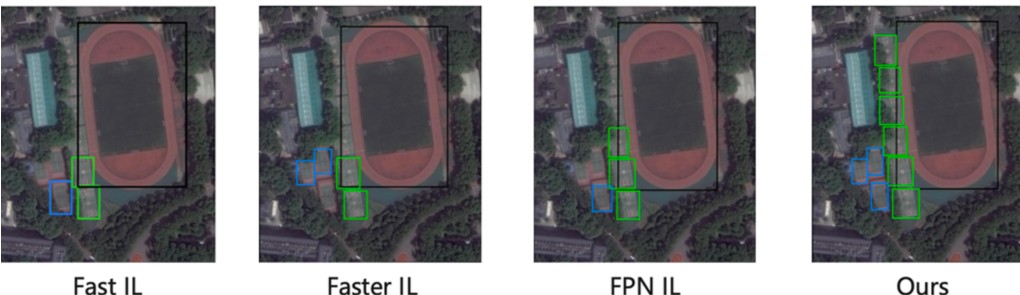

Fast IL    Faster IL    FPN IL    Ours

**Figure 3  The visualization detection results of the proposed method on the DOTA dataset.**

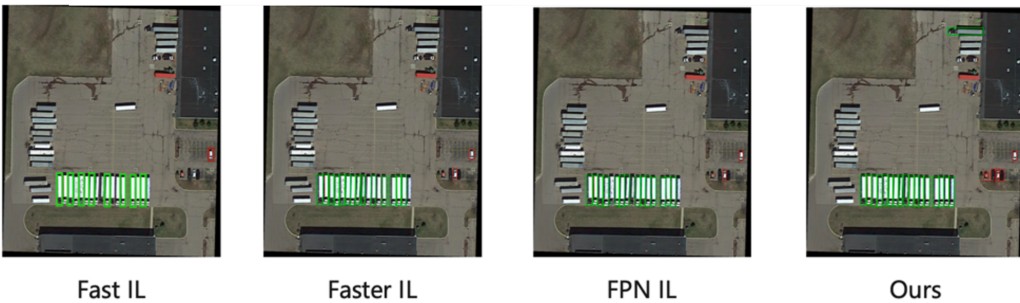

Fast IL    Faster IL    FPN IL    Ours

**Figure 4  The visualization detection results of the proposed method on the DIOR dataset.**

method can learn information about the similarity between classes, preventing catastrophic forgetting while accelerating the learning of new classes.

Figure 4 shows the comparison of the visualization results in the DIOR dataset with low similarity of learning tasks, and since the proposed method can adjust the distillation weights adaptively according to the task similarity, it can also obtain better detection results.

Furthermore, the heatmaps are used to verify the effectiveness of the similarity distillation method we proposed in Fig. 5. In the heatmaps, the darker the color of the heat map, the more critical the area is. Figure 1 shows that we first learn the class SBF and then learn the class BC. From the change in the heat map of the network, the SBF in the bottom right corner of the heatmap (a) is activated. When the network continues to learn the class BC, both areas can be activated, which shows that the proposed incremental learning method can remember the previous knowledge well. Moreover, after learning BC, the activation area of SBF changes from the annular to the central square area, which shows that the network can learn the similarity features between classes.

Based on the public natural scene image dataset VOC, we tested the class similarity distillation method to verify the effectiveness of class incremental object detection, as shown in Table 3. For CSD in the last row, we used the settings described in the implementation details. To compare, we also replaced the CSD loss with the L2 loss to minimize the distance between the selected features. As a result of the performance of CSD on average accuracy,

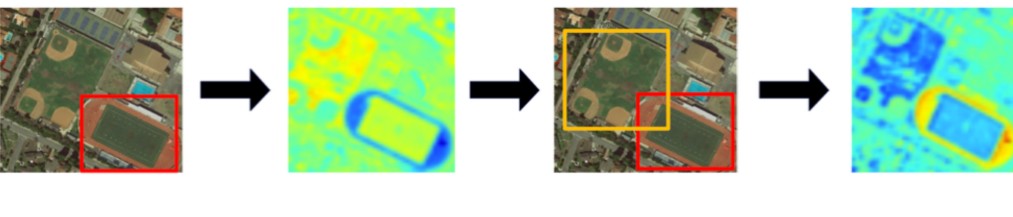

Class SBF                    Incremental Class BD

**Figure 5** **The heatmaps to verify the effectiveness of the proposed similarity distillation.**

**Table 3** **Testing the class similarity distillation way to verify the effectiveness for class incremental object detection.**

| Methods | A | B | C | D | AA |
|---|---|---|---|---|---|
| Baseline | 48.75 | – | – | – | 48.75 |
| Baseline | 44.12 | 58.34 | | – | 51.21 |
| Baseline | 30.77 | 33.56 | 56.24 | – | 40.87 |
| Baseline | 15.33 | 18.25 | 43.28 | 35.66 | 28.54 |
| Ours | 48.75 | – | – | – | 48.75 |
| Ours | 45.25 | 57.88 | – | – | 52.24 |
| Ours | 31.57 | 34.12 | 57.26 | – | 41.23 |
| Ours | 16.22 | 19.23 | 44.34 | 36.66 | 29.13 |

it is consistently superior to other methods, proving that it is more appropriate to obtain a trade-off between stability and plasticity for continuous object detection by using CSD. For 19+1 and 15+5 tasks, CSD is more effective than the L2 loss on average accuracy. Since CSD enforces the instance-level features of the incremental model to imitate the features of the old incremental model to a high degree, the performance of the old classes can be adequately maintained.

In contrast, the performance of the new classes will be suppressed at the same time. A comparison of CSD and L2 loss on average accuracy shows that CSD is more effective than L2 loss for 19+1 and 15+5 tasks. CSD enforces instance-level features of the incremental model to entirely mimic those of the old model so that the performance of old classes can be maintained simultaneously as the performance of new classes is suppressed simultaneously.

## Ablation study
An ablation study is performed to validate the contribution of distillation loss in the DOTA dataset. Like the experiment in Table 2, we incrementally learn the following seven classes. The results of the ablation experiments in Table 4 show the effectiveness of the proposed CSD and GSD. In Table 4, the second column is the result obtained without the distillation algorithm, the second and third columns are the AA obtained by using one distillation loss, respectively, and the last column is the result of using two distillation losses at the same time. Each distillation loss we proposed can boost AA, and the best results can be obtained when used together.

**Table 4  Ablation study is performed to validate the contribution of distillation loss in the DOTA dataset.**

| Module | 1 | 2 | 3 | 4 |
|---|---|---|---|---|
| CSD | – | ✓ | – | ✓ |
| GSD | – | – | ✓ | ✓ |
| AA | 14.5 | 65.7 | 64.3 | 66.6 |

## DISCUSSION

Despite the promising gains that can be achieved with our proposed class similarity distillation (CSD) and global similarity distillation (GSD) for class incremental object detection in remote sensing, there are still several concerns that need to be further researched in the future. First, there is a significant discrepancy between the outcomes of sequential addition training and the outcomes of joint training in all classes, which may be caused by the gradual accumulation of mistakes during the incremental learning process. Additionally, the chosen features for correlation distillation need to be more accurate after numerous learning stages. Due to the lack of data and the trade-off between stability and plasticity, the performance of both old and new classes cannot be improved simultaneously.

## CONCLUSION

In this article, we propose a novel class similarity distillation-based class incremental object detection method in remote sensing images that considers the similarity of new and old classes. First, class similarity distillation (CSD) was proposed to determine the plasticity and stability of the model during local distillation in the backbone of the object detector. To further mitigate catastrophic forgetting of the incremental model, we also introduced a global similarity distillation (GSD) loss to maximize the mutual information between old and new classes. Results on DOTA, DIOR, and VOC datasets demonstrate that the proposed method is effective in incremental class learning to detect objects in remote sensing images without forgetting what has previously been learned.

In the future, it will be possible to combine incremental object detection with other techniques, such as those found in *Morioka & Hyvarinen (2023)*, to maintain better feature discrimination within the incremental class procedure. We will also consider designing novel methods for classifiers and regressors to further boost class incremental object detection performance.

### Funding
The authors received no funding for this work.

### Competing Interests
The authors declare there are no competing interests.

## Author Contributions

- Mingge Shen conceived and designed the experiments, analyzed the data, authored or reviewed drafts of the article, and approved the final draft.
- Dehu Chen conceived and designed the experiments, performed the experiments, analyzed the data, performed the computation work, prepared figures and/or tables, authored or reviewed drafts of the article, and approved the final draft.
- Silan Hu analyzed the data, prepared figures and/or tables, authored or reviewed drafts of the article, and approved the final draft.
- Gang Xu performed the experiments, performed the computation work, prepared figures and/or tables, and approved the final draft.

## Data Availability

The DOTA dataset is available at https://captain-whu.github.io/DOTA/dataset.html.

Ding J, Xue N, Xia G-S, Bai, X, Yang, W, Yang, M, Belongie S, Luo J, Datcu M, Pelillo M, Zhang L. 2021. Object Detection in Aerial Images: A Large-Scale Benchmark and Challenges. IEEE Transactions on Pattern Analysis and Machine Intelligence. DOI 10.1109/TPAMI.2021.3117983

The DIOR dataset is available at: Li K, Wan G, Cheng G, Meng L, Han J. 2020. Object detection in optical remote sensing images: A survey and a new benchmark. ISPRS journal of photogrammetry and remote sensing. 159:296-307.

## Supplemental Information

Supplemental information for this article can be found online at http://dx.doi.org/10.7717/peerj-cs.1583#supplemental-information.

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
