# Peer review of "Class incremental learning of remote sensing images based on class similarity distillation"

_PeerJ Computer Science, doi:10.7717/peerj-cs.1583_

## Round 0.1 · original submission · Major Revisions

Please make sure that the revised manuscript addresses the reviewer comments.

Reviewer 1 ·

Basic reporting

The paper proposes an incremental learning method for remote sensing image target recognition, which is of great practical value. However, the English writing in the paper is very poor, which significantly hinders understanding. In addition, the paper contains many non-standard elements, such as improper referencing of the literature, unnumbered equations, and unexplained symbols in the equations.

I suggest that the authors should thoroughly rewrite the entire paper, or even use ChatGPT to perform language polishing for the paper.

Experimental design

The comparative experiments are not sufficient, and it is recommended to compare with the latest methods.

In addition, the dataset used should be properly cited with the corresponding references.

Validity of the findings

The proposed method is somewhat innovative.

Reviewer 2 ·

Basic reporting

The equations are not referenced.
Some terms need to be defined.
There are ambiguities in the notation (e.g. i , C, ...)
(see the section bellow for Additional comments)

Experimental design

Not very original, distillation of class information using cosine similarity from exemplars was already proposed in
Hou & al. "Learning a Unified Classifier Incrementally via Rebalancing", CVPR2019.

Validity of the findings

no comment

Additional comments

This paper, entitled “Class Incremental Learning of Remote Sensing Images Based on Class Similarity Distillation” , presents a method for incremental object detection. The algorithm that introduces the information similarity uses RPN as a baseline.
The document has several serious shortcomings, which I will identify in the following points:
Q1 : In line 200, Z is defined as R^{HWC} whereas in line 204 M_t is denoted as R^{HxWC}. What is the difference between the two notations HxWC and HWC?

Q2: equation 1 does not refer to the number of channels C?

Q 3: The denominator [i:y_i:c] should be clearly defined. The notation is not explicit.

Q4: “i” is used everywhere, as a pixel index, as a batch index which makes understanding the process more difficult to follow.

Q5: Each equation referred to in the text should be identified by the corresponding number. Again, this complicates understanding for the readers.

Q6 : line 215, “new” is duplicated ?

Q7: The approach is based on the RPN which means that the joint training of the FPN-il and the CSD should be similar in terms of AA. However, in table 2, as we can see the AA is identical for BC, ST, TR, and on the other hand it is always better for SFB, Harbour, SP, and HC? a clear argument should be given to clarify these cases.

Q8: line 31, Fig.1 should be fig 5 !!

Q9: line 301, what do the SBF and BD classes stand for?

Q10: line 360, citation guidelines should be removed!

---

## Round 0.2 · Minor Revisions

It would be beneficial, if the authors could take a clear stance regarding the CSD method proposed in Feng 2021 and compare their proposed approach with this similar CSD-based method. See comment from Reviewer 2.

Reviewer 1 ·

Basic reporting

The manuscript have been greatly improved. I have no more comments.

Experimental design

The manuscript have been greatly improved. I have no more comments.

Validity of the findings

The manuscript have been greatly improved. I have no more comments.

Additional comments

The manuscript have been greatly improved. I have no more comments.

Reviewer 2 ·

Basic reporting

The authors have provided responses to all of my inquiries.

There are still some English writing errors

Experimental design

no comment

Validity of the findings

Not very original as CSD was investigated in many papers.

Additional comments

A Category-wise similarity distillation (CSD) was also proposed by Feng 2021, in "Double Similarity Distillation for Semantic Image Segmentation".
The authors should compare and cite this paper.


There are still some English writing errors (e.g. "given a object detector" should be "given an object detector")

---

## Round 0.3 · accepted · Accept

From the submitted revision and related rebuttal letter, I am satisfied with the author's efforts to address reviewer comments.